# Mitochondrial Genomes of Two *Thaparocleidus* Species (Platyhelminthes: Monogenea) Reveal the First rRNA Gene Rearrangement among the Neodermata

**DOI:** 10.3390/ijms20174214

**Published:** 2019-08-28

**Authors:** Dong Zhang, Hong Zou, Ivan Jakovlić, Shan G. Wu, Ming Li, Jin Zhang, Rong Chen, Wen X. Li, Gui T. Wang

**Affiliations:** 1Key Laboratory of Aquaculture Disease Control, Ministry of Agriculture and State Key Laboratory of Freshwater Ecology and Biotechnology, Institute of Hydrobiology, Chinese Academy of Sciences, Wuhan 430072, China; 2University of Chinese Academy of Sciences, Beijing 101408, China; 3Bio-Transduction Lab, Biolake, Wuhan 430075, China

**Keywords:** dactylogyridae, ancyrocephalinae, phylogenetics, paraphyly, ancylodiscoididae, large non-coding region

## Abstract

Phylogenetic framework for the closely related Ancylodiscoidinae and Ancyrocephalinae subfamilies remains contentious. As this issue was never studied using a large molecular marker, we sequenced the first two Ancylodiscoidinae mitogenomes: *Thaparocleidus asoti* and *Thaparocleidus varicus*. Both mitogenomes had two non-coding regions (NCRs) that contained a number of repetitive hairpin-forming elements (RHE). Due to these, the mitogenome of *T. asoti* (16,074 bp) is the longest among the Monogenea; especially large is its major NCR, with 3500 bp, approximately 1500 bp of which could not be sequenced (thus, the total mitogenome size is ≈ 17,600 bp). Although RHEs have been identified in other monopisthocotyleans, they appear to be independently derived in different taxa. The presence of RHEs may have contributed to the high gene order rearrangement rate observed in the two mitogenomes, including the first report of a transposition of rRNA genes within the Neodermata. Phylogenetic analyses using mitogenomic dataset produced Dactylogyrinae embedded within the Ancyrocephalinae (paraphyly), whereas Ancylodiscoidinae formed a sister-group with them. This was also supported by the gene order analysis. *28S*
*rDNA* dataset produced polyphyletic Dactylogyridae and Ancyrocephalinae. The phylogeny of the two subfamilies shall have to be further evaluated with more data.

## 1. Background

Species from the genus *Thaparocleidus* (Ancylodiscoidinae or Ancylodiscoididae, see below) are common monogenean parasites found in catfishes, but their phylogeny remains debated. Half a century ago, in Bychowsky’s classification [1], this genus (and other genera nowadays putatively assigned to Ancylodiscoidinae) was originally assigned to the subfamily Ancyrocephalinae, family Dactylogyridae. Price [2] and Gusev [3] later assigned some of the genera from Ancyrocephalinae to three new subfamilies within the Dactylogyridae: Ancylodiscoidinae, Heteronchocleidinae and Anacanthorinae. Later, Bychowsky and Nagibina [4] excluded Ancyrocephalinae from the family Dactylogyridae, and gave it the family status (Ancyrocephalidae). They also assigned Ancylodiscoidinae into this newly established family. However, Kritsky and Boeger [5] found that Ancyrocephalidae was not a monophyletic group, and suggested placing all subfamilies within this family back into the family Dactylogyridae, as well as reverting the Ancyrocephalidae back to the subfamily status (within the Dactylogyridae). Finally, Lim, et al. [6] proposed to elevate the Ancylodiscoidinae to the family status. These incompatible classifications demonstrate the deeply complex and unresolved relationship of Ancylodiscoidinae and Ancyrocephalinae. However, as all of the above studies were based on morphological data, several studies later attempted to resolve the phylogenetic relationships within the family Dactylogyridae using molecular data: *18S rDNA* [7], *28S rDNA* [8], and a combination of *18S*, *28S* and *16S rDNA* [9]. In these studies, predominantly marine Ancyrocephalinae species (with a few freshwater outliers, see clade B in [8]) constantly clustered with the Dactylogyrinae and Pseudodactylogyrinae species (M-clade henceforth). However, Ancylodiscoidinae either formed a sister-clade with freshwater Ancyrocephalinae species [7,9,10], or grouped with Ancyrocephalinae species that parasitize on Siluriformes (AA-clade, henceforth) [8]. In the latter results, M-clade formed a sister-clade with AA-clade (but with low support), whereas freshwater Ancyrocephalinae were basal to them, i.e., formed a sister-clade with the former two groups. Therefore, both morphological and molecular studies failed to obtain monophyletic Ancyrocephalinae, and the relationship of Ancylodiscoidinae and Ancyrocephalinae remains unresolved.

Several previous studies suggest that morphological characters are a poor phylogenetic marker in many microscopic parasitic animals, often exhibiting host-specific morphological variability [11,12,13]. It is likely that at least part of the underlying cause for the multiple incongruent hypotheses inferred using morphological data. Small molecular markers often also have limited resolving power [14], but relationships of Ancylodiscoidinae and Ancyrocephalinae have not been studied using a large molecular marker (previous molecular studies employed 1 to 3 genes), comprised of multiple concatenated genes, due to unavailability of suitable data. Although such markers can also produce conflicting and homoplastic signals [15,16], and despite the fact that there may not exist such a thing as the ‘accurate’ phylogenetic tree [17], the high resolution that they carry may help us shed some light on the contentious relationships of Ancylodiscoidinae and Ancyrocephalinae. Due to a number of peculiarities that make them a suitable candidate for such tasks, mitochondrial genomes (mitogenomes) have become a popular tool in population genetics [18], phylogenetics [19,20] and diagnostics [21] studies.

Among the monogenean mitogenomes available in the GenBank, Ancyrocephalinae are relatively well-represented with seven mitogenomes, but Ancylodiscoidinae currently remain unrepresented. This scarcity of data has thus far hampered studies of the phylogeny of Ancylodiscoidinae and Ancyrocephalinae subfamilies from the mitogenomic perspective. To address this, we sequenced and characterized the mitogenomes of two Ancylodiscoidinae species: *Thaparocleidus asoti* and *Thaparocleidus varicus*. Here, we used the newly sequenced mitogenomes and the data available from public databases to investigate phylogenetic relationships of the two subfamilies, and their position within the family Dactylogyridae.

## 2. Results

### 2.1. Genome Organization and Base Composition

The complete mitogenome of *Thaparocleidus varicus* (MN151339) was 14,088 bp in size, and the nearly complete mitogenome of *Thaparocleidus asoti* (MN151340) was 16,074 bp in size (Figure 1). Both mitogenomes contain the standard [22] 36 flatworm mitochondrial genes, including 12 protein-encoding genes (PCGs; *atp8* is absent), 22 tRNA genes and two rRNA genes (Table 1 and Figure 1). Majority of PCGs of the two studied mitogenomes used standard initial codons for the genetic code 9 (echinoderm and flatworm mitochondrion): ATG or GTG. However, it proved difficult to determine the initial codons of the *nad4* and *cox1* genes in *T. varicus*. Canonical stop codons for the genetic code 9 (echinoderm and flatworm mitochondrion), TAA and TAG, were found in all 12 PCGs, except for *cox2* in *T. asoti*, which used the abbreviated T-- codon (Table 1). The architecture and similarity of orthologous sequences for the two studied mitogenomes are summarized in Table 1. Average sequence similarity of PCGs between the two studied *Thaparocleidus* mitogenomes ranged from 68.89% (*nad4L*) to 84.22% (*cytb*) (Table 1). We also investigated the codon usage, RSCU, and codon family (corresponding to the amino acids) proportions between the two *Thaparocleidus* species (Appendix A). Leu2, Phe, Ile and Val were the most common codon families, predominantly encoded by adenosine and thymine-rich codons, such as TTA in Leu2, TTT in Phe, ATT and ATA in Ile (Appendix A).

### 2.2. Non-Coding Regions

Two large non-coding regions (NCR1 and NCR2) were found in both mitogenomes (Figure 2). NCR1 was located between *nad5* and *trnK* genes, whereas NCR2 was positioned between *trnG* and *cox3* (Figure 2). NCR1 of *T. varicus* was 478 bp in size. We managed to sequence only 2020 bp of the NCR1 of *T. asoti*, but we have successfully amplified it and estimated its size to be approximately 3500 bp. Therefore, we estimate that around 1480 bp remains unsequenced. NCR2 was 792 bp and 416 bp in size in *T. asoti* and *T. varicus,* respectively.

All the NCRs contained highly repetitive regions (HRR). As the sequencing gap of *T. asoti* was located in the central of HRR of the NCR1, we hypothesized that the gap was probably composed of tandem repeats (TRs). In this way, counting the unsequenced gap, the HRR of NCR1 in *T. asoti* probably contained 23 uninterrupted TRs, assuming identical repeat units (132 bp, Figure 2). The HRR of NCR2 in *T. asoti* was comprised of 11 uninterrupted TRs, where repeat units 2–11 were identical (63 bp), whereas unit 1 exhibited one nucleotide mutation at the tenth position (Figure 2). Similarly, HRR of NCR1 in *T. varicus* was composed of three 166 bp-long TRs, where repeat unit 3 was severely truncated to only 53 bp (lost 113 nucleotides at the 3′ end), and had a nucleotide mutation at the seventh, eighth and 40th positions (Figure 2). The HRR of NCR2 in *T. varicus* was composed of five TRs, where repeat units 1–4 were identical (48 bp), whereas unit 5 only contained the first 13 nucleotides of the TR, and it exhibited one A to C mutation at the third position and one nucleotide deletion at the fifth position (Figure 2). The consensus repeat patterns of all HRRs in *T. asoti* and *T. varicus* are capable of forming double to hexa stem-loop structures (Figure 2).

### 2.3. Phylogeny

Regardless of the method used, Bayesian inference (BI) and maximum likelihood (ML) produced identical topologies (Figure 3). Gyrodactylidea was placed at the base of Monopisthocotylea, whereas the rest of monopisthocotyleans were split into two clades: Tetraonchidea and Dactylogyridea + Capsalidea. Most of the nodes exhibited high support values, except for some internal nodes of Dactylogyridae (Figure 3). Despite these low support values, among the three Dactylogyridae subfamilies, Ancyrocephalinae was rendered paraphyletic by the embedded Dactylogyrinae, whereas Ancylodiscoidinae formed a sister-group with them, with maximum support values (BI/ML = 1/100) (Figure 3). *28S rDNA* data also failed to resolve the Ancyrocephalinae/Ancylodiscoidinae debate, as both BI and ML analyses produced instable topologies, with polyphyletic Dactylogyridae and Ancyrocephalinae, and Ancylodiscoidinae embedded within the Ancyrocephalinae (Appendix A).

### 2.4. Gene Orders

The gene orders (GO) of the two studied *Thaparocleidus* species are identical, but notably different from other monogeneans, exhibiting many tRNA and rRNA genes’ rearrangements (Figure 3). However, the GO between *trnG* to *trnI* is conserved. The high rate of GO rearrangements in the two *Thaparocleidus* species was further corroborated by the low similarity values produced by pairwise comparisons with other monogeneans: values ranged from 102 (compared with *Paratetraonchoides inermis*) to 326 (compared with *Cichlidogyrus sclerosus*), where the value of 1254 indicates identical GOs (Table 2).

## 3. Discussion

Despite the sequencing gap in the non-coding region, the mitogenome of *T. asoti* is the longest monogenean mitogenome reported so far. Unlike *T. varicus* and most other monopisthocotyleans, which have an overlap between *nad4L* and *nad4* genes, *T. asoti* had an 87 bp gap between them (Appendix A). The A+T content of the two *Thaparocleidus* species was relatively high among the 33 selected monogeneans (Appendix A), and it was notably higher than in other dactylogyrids (Figure 4). The AT skewness of the two *Thaparocleidus* species was similar to other dactylogyrids, except for *Tetrancistrum nebulosi*, *Ancyrocephalus mogurndae* and *Euryhaliotrema johnii*, which were outliers, with a somewhat lesser magnitude of (negative AT) skews (Figure 4). On the basis of results reported in other related species (Appendix A), as a working hypothesis, we propose TTG as the initial codon of *nad4*, and ATT as the start codon of *cox1*. TTG was proposed as an alternative start codon for flatworm mitogenomes before [23]. Noteworthy, all codons from the four prominent codon families used thymine in the second position. In addition, the second position of the PCGs exhibited the highest negative AT skewness (i.e., T preference) in comparison to other mitogenomic elements (Figure 4). This is probably a reflection of the fact that codons for hydrophobic amino acid residues, which are functionally preferred for conformational stability of mitochondrial proteins, mostly have T in the second codon position [24].

The size of the NCRs in *T. asoti* was much larger than that in *T. varicus*, resulting in approximately 25% larger mitochondrial genome in this species (Table 3). Repetitive stem-loop elements are not uncommon within the subclass Monopisthocotylea; they were also found in *Dactylogyrus lamellatus* [25], diplectanids [26] and *Tetraonchus monenteron* [27]. However, as these species are phylogenetically distant and secondary structures and nucleotide composition of the stem-loop elements were largely different among different species, this suggests multiple independent invasions [28] of these features. These findings consistently reject the hypothesis that monopisthocotylids possess fewer and smaller (in size) TRs in the LNCR than polyopisthocotylids [29]. Since NCRs with repetitive features are believed to indicate control regions [28], and the presence of tandem repeats forming stable secondary structure is often associated with the initiation of replication in mitochondria [20,30,31], it appears likely that these repeat regions are embedded within the control region. Given that the non-sequenced gap in *T. asoti* was located within the TR region, this would also explain why we failed to sequence this segment, as it is likely that HRRs in this segment formed complex structures that interfered with the sequencing [32,33].

The ordinal relationships obtained in this study were similar to the topology obtained in an earlier mitogenomic study [26]. Notably, Capsalidea was embedded within the Dactylogyridea order, thus causing paraphyly of the latter order (Figure 3), which was discussed before [26]. With regards to the three Dactylogyridae subfamilies, the results suggest a closer relationship between Ancyrocephalinae and Dactylogyrinae than between Ancylodiscoidinae and Ancyrocephalinae subfamilies, which contradicts some of the morphology-based hypotheses that grouped the species of Ancylodiscoidinae within the Ancyrocephalid(n)ae (sub)family [1,4]. Beyond the *Thaparocleidus* species and the mesoparasitic *Enterogyrus malmbergi*, the rest of dactylogyrids belonged to the M-clade (see Background section) [8,10]. However, the freshwater *A. mogurndae* (Ancyrocephalinae) was embedded within a clade that contained marine *E. johnii* and *T. nebulosi*, and freshwater *Cichlidogyrus mbirizei*, *C. sclerosus* and *C. halli*, thus contradicting previous molecular studies, which placed *A. mogurndae* within the clade containing Pseudodactylogyrinae and Dactylogyrinae [8,10]. However, due to the limited taxon sampling (only 1/7 representatives of the speciose Dactylogyrinae/Ancyrocephalinae subfamilies were available, respectively; and only three of the nine subfamilies within the Dactylogyridae were represented), we cannot infer the relationships of the three subfamilies (Ancylodiscoidinae, Ancyrocephalinae and Dactylogyrinae) with confidence. The instable result indicates that *28S rDNA* has too low a resolution to resolve the phylogeny of Dactylogyridae. However, the closer relationship between Ancyrocephalinae and Dactylogyrinae than between Ancylodiscoidinae and Ancyrocephalinae was also supported by our gene order analysis.

The rearrangement of rRNA genes is the first reported within the subphylum Neodermata. In all other neodermatans, these two genes are located between *cox1* and *cox2* genes [34], but in the two newly-sequenced *Thaparocleidus* species, they are translocated to the position between *nad5* and *cox3*, together with several tRNA genes (Figure 3). Noteworthy, in dactylogyrideans, tetraonchideans (Figure 3) and cestodes [35], the major NCR is usually found in the region between *nad5* and *cox3*. As we hypothesized that the NCRs of these two *Thaparocleidus* species harbor the control regions (see “Non-coding regions” section), our results are in agreement with the hypothesis that genes adjacent to the control region exhibit higher rates of rearrangements [36]. This is probably associated with the fact that hairpin elements can facilitate recombination and rearrangement events in the mitogenome [28]. Noteworthy, the GO most similar to that of *Thaparocleidus* was the putative ancestral neodermatan GO (AN-GO henceforth) [34]. According to the hypothesis proposed by Zhang, et al. [34], the GO of the common ancestor of all Dactylogyridae species most probably possessed the AN-GO, as this gene order was probably retained throughout all of the common ancestors leading to the extant species possessing the AN-GO in the Dactylogyridae clade: *C. sclerosus*, *T. nebulosi* and *Enterogyrus malmbergi* (Figure 3). This is also supported by the hypothesis proposed by Boore [37]: GOs are unlikely to revert to a primitive condition.

The highly rearranged GO of the two *Thaparocleidus* species may have uncovered a new group of monopisthocotylean monogeneans that exhibit fast-evolving GOs. Other monopisthocotyleans that exhibit elevated mitogenomic GO rearrangement rates include diplectanids [26], tetraonchideans [11] and *Aglaiogyrodactylus forficulatus* within the gyrodactylids [38] (Figure 3 and Table 2). However, the GOs of these groups of species exhibit low mutual similarity (Table 2), which indicates that all of these accelerations of GO evolution occurred independently, and share few common rearrangement patterns. This consistently confirms the hypothesis that evolution of mitogenomic GO arrangements is discontinuous in monogeneans [11,26], as GOs in a proportion of monogenean taxa are highly variable, whereas the remaining are conserved (Table 2 and Figure 3). Sequencing of future mitogenomes shall show whether the GO pattern exhibited by these two *Thaparocleidus* species may represent the synapomorphic arrangement of the subfamily Ancylodiscoidinae, and whether the GOs can be used to resolve some of the taxonomic and phylogenetic debates discussed herein. For example, the GO analysis supported the phylogenetic results that separated Ancylodiscoidinae from the two closely related Ancyrocephalinae and Dactylogyrinae (Figure 3). However, GOs should be used with utmost caution for phylogenetic purposes, as the discontinuity in GO rearrangements in monogeneans might produce misleading evolutionary signals and cause long-branch attraction artifacts [34].

## 4. Materials and methods

### 4.1. Specimen Collection and Identification

*Thaparocleidus asoti* and *T. varicus* were obtained from the gills of a single *Silurus meridionalis* (Chen, 1977) (Siluriformes: Siluridae) specimen, bought at a local market in the Wuhan city, Hubei Province on 6 May, 2017. They were morphologically identified by the hard parts of the haptor and reproductive organs as described in Wu, et al. [39]. Additionally, to confirm the taxonomic identity from the molecular perspective, their *28S rRNA* genes were amplified using the C1 (5′-ACCCGCTGAATTTAAGCAT-3′) and D2 (5′-TGGTCCGTGTTTCAAGAC-3′) primer pair [40]. Both species share a very high identity with corresponding conspecific homologs available in the GenBank: 99.35% (762/767 bp) for *T. varicus* (DQ157668), and 100% (800/800 bp) for *T. asoti* (MG601546). All sampled and identified parasites were first washed in 0.6% saline and then stored in 100% ethanol at 4 °C.

### 4.2. DNA Extraction, Amplification and Sequencing

To ensure a sufficient amount of DNA for amplification and sequencing of these small parasites, we used two types of DNA for amplification and sequencing: mixture DNA (extracted from 20 specimens) and individual DNA (a single specimen). Both were extracted using TIANamp MicroDNA Kit (Tiangen Biotech, Beijing, China). First, we selected 14 monogenean mitogenomes from GenBank, aligned them using ClustalX [41], designed degenerate primer pairs (Appendix A) matching the generally conserved regions of mitochondrial genes, and amplified the whole mitogenome using the mixture DNA. Specific primers, based on these obtained fragments, were then designed using Primer Premier 5 [42], and the remaining mitogenome was amplified and sequenced in several PCR steps. PCR products were sequenced bi-directionally using both degenerate and specific primers mentioned above on an ABI 3730 automatic sequencer (Sanger sequencing). All obtained fragments were BLASTed [43] to confirm that the amplicon is the actual target sequence. We carefully examined the chromatograms, paying close attention to double peaks or any other sign of the existence of two different sequences. To address the possibility of intraspecific sequence variation present in the mixture DNA, we then used individual DNA and long-range PCR to re-sequence the mitogneomes. If the two sequences differed, we used the DNA extracted from a single specimen to assemble the final mitogenome, thereby ensuring that each sequence belongs to a single specimen.

### 4.3. Sequence Annotation and Analyses

Both mitogenomes were assembled and annotated following the procedure described before [11,25,35,44] using DNAstar v7.1 software [45], MITOS [46] and ARWEN [47] web tools: after assembling with the help of DNAstar, MITOS was used to annotate the mitogenome, Protein-coding genes (PCGs) were determined by searching for ORFs using genetic code 9 (echinoderm and flatworm mitochondrion) and aligning with homologs, two rRNA genes were also confirmed by the alignment with homologs, and tRNAs were identified by combining the results of ARWEN and MITOS. An in-house PhyloSuite software [48] was used to parse and extract the annotations recorded in Word documents, as well as create GenBank submission files and organization tables for mitogenomes. The same software was used to make genomic statistics of the mitogenome of monogeneans. Codon usage, amino acid proportion and relative synonymous codon usage (RSCU) for 12 protein-encoding genes (PCGs) of the two studied *Thaparocleidus* species were calculated and sorted using PhyloSuite, and finally the RSCU figure drawn using ggplot2 [49] plugin. Hierarchical clustering and heatmap analyses were drawn using the ComplexHeatmap package [50] implemented in R, with the help of the statistics file generated by PhyloSuite. Tandem Repeats Finder [51] was invoked to find tandem repeats in the non-coding regions, and their secondary structures were predicted by Mfold software [52]. Genetic distances (identity) among mitogenomic sequences were computed with the “DistanceCalculator” function in Biopython [53] using “identity” model.

### 4.4. Phylogenetic Analyses

Phylogenetic analyses were conducted using the two newly sequenced *Thaparocleidus* mitogenomes and 31 monogenean mitogenomes available in the GenBank (5/7/2019). Six polyopisthocotylid monogeneans were used as outgroups (Appendix A). We used a dataset comprised of concatenated amino acid sequences of all 12 protein-coding genes for the phylogenetic analysis. Additionally, to get a nuclear perspective on the topic, a *28S* gene dataset that closely matched the taxonomic composition of the mitogenomic dataset was also used to conduct phylogenetic analyses (Appendix A). ModelFinder [54] plugin integrated into PhyloSuite was used to calculate the Best-fit model. mtZOA+F+I+G4 was selected as the optimal model for the mitogenomic dataset, whereas GTR+F+G4 was chosen for the *28S* dataset. Phylogenetic analyses were performed using two different algorithms: ML and BI. ML analysis for both datasets was carried out in RAxML [55] using a ML+rapid bootstrap (BS) algorithm with 1000 replicates. Bayesian inference with GTR+F+G4 model for *28S* dataset was conducted in MrBayes 3.2.6 [56] plugin in PhyloSuite. MrBayes was run with default settings, and 5 × 10^6^ metropolis-coupled MCMC generations. Stationarity was considered to be reached when the average standard deviation of split frequencies was < 0.01, ESS (estimated sample size) value > 200, and PSRF (potential scale reduction factor) approached 1. Bayesian inference analyses for amino acid dataset were conducted using the empirical MTZOA model and PhyloBayes (PB) MPI 1.5a [57]. For each analysis, two MCMC chains were run after the removal of invariable sites from the alignment, and the analysis was stopped when the conditions considered to indicate a good run (PhyloBayes manual) were reached: maxdiff < 0.1 and minimum effective size > 300. Non-coding regions of the selected monopisthocotyleans were identified and extracted from GenBank files using PhyloSuite, with the threshold set at 200 bp. iTOL dataset files produced by PhyloSuite were then used to visualize and annotate the phylograms and gene orders in iTOL [58].

## 5. Conclusions

Mitogenomes of both *T. asoti* and *T. varicus* contain two large non-coding regions, which were comprised of a number of repetitive hairpin-forming elements (RHE). The number of repeats varied between the two species, resulting in the exceptionally large genome of *T. asoti* (although incomplete), the largest among all available monogeneans. The gene order exhibited by both species was notably different from other monogeneans, with the first rearrangement of rRNA genes reported among the subphylum Neodermata thus far. Ancyrocephalinae and Dactylogyrinae were closely related in the phylogenetic results using mitogenome dataset, whereas Ancylodiscoidinae formed a sister-group with them. This relationship was also supported by the gene order. *28S rDNA*-based analyses failed to produce monophyletic Ancyrocephalinae and Dactylogyridae. Our phylogenetic results inferred using mitogenomic dataset contradict previous phylogenetic studies (morphology and molecular marker-based). Limited availability of mitogenomes (only three of nine subfamilies of Dactylogyridae were available, and some with too few representatives) and weakly supported topology prevent us from making conclusions with confidence. Sequencing of additional molecular data, such as mitogenomes, transcriptomes or multiple nuclear genes, will be needed to resolve the interrelationships of Dactylogyridae.

## Figures and Tables

**Figure 1 ijms-20-04214-f001:**
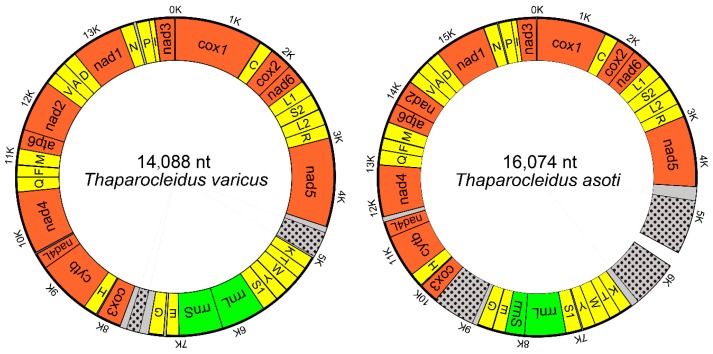
Maps of the circular mitochondrial genomes of *Thaparocleidus asoti* and *Thaparocleidus varicus*. Protein-coding genes are red, tRNAs are yellow, rRNAs are green, and non-coding regions (NCR) are grey. The location of the highly repetitive regions (HRR) within NCR are shown with black spot. The unsequenced gap is shown in the in the NCR of *T. asoti*.

**Figure 2 ijms-20-04214-f002:**
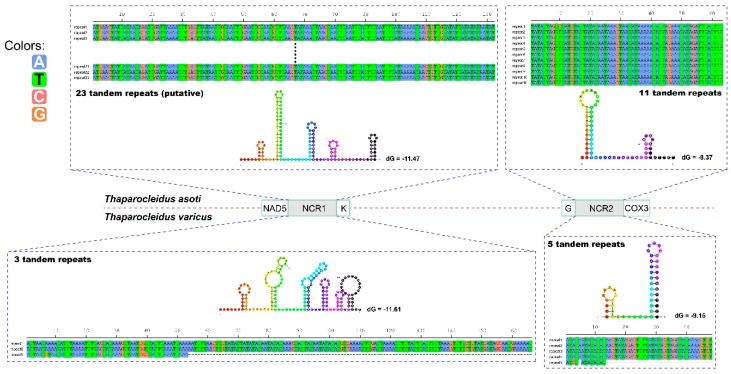
Stem-loop structures of the consensus repeat patterns in repetitive regions of the non-coding regions of *Thaparocleidus asoti* and *Thaparocleidus varicus*. Thermodynamic energy values (dG) are shown next to the secondary structures. The nucleotide A is blue, T is green, C is deep pink and D is orange.

**Figure 3 ijms-20-04214-f003:**
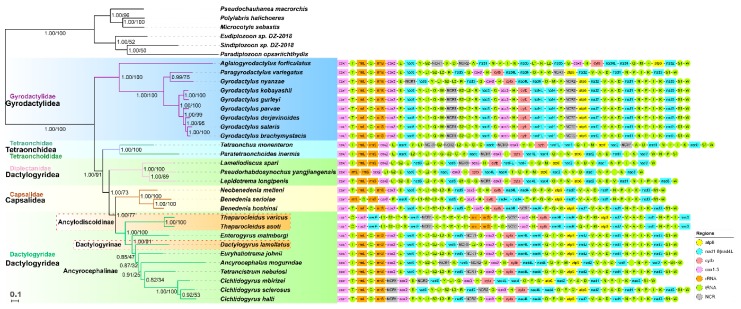
A phylogram reconstructed using mitogenomes of 33 monogeneans and the mtZOA model. Scale bar corresponds to the estimated number of substitutions per site. Statistical support values of Bayesian analyses and maximum likelihood are shown by the nodes (left/right, respectively). Taxonomic families and orders are shown in different colors. Gene orders and non-coding regions (grey boxes) are shown to the right of the tree.

**Figure 4 ijms-20-04214-f004:**
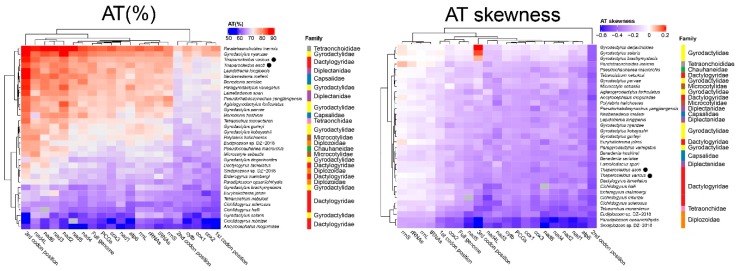
Hierarchical clustering of the A+T content and AT-skewness of various mitogenomic elements among the monogenean mitogenomes.

**Table 1 ijms-20-04214-t001:** Comparison of the annotated mitochondrial genomes of *Thaparocleidus asoti* and *Thaparocleidus varicus*. Negative values in the ‘Intergenic Nucleotides’ column indicate overlaps.

Gene	Position		Size	Intergenic Nucleotides	Codon		Anti-codon	Identity
	From	To			Start	Stop		
*Thaparocleidus asoti*/*Thaparocleidus varicus*								
*cox1*	1/1	1554/1557	1554/1557		ATG/ATT	TAG/TAG		82.59
*trnC*	1554/1566	1617/1630	64/65	−1/8			GCA/GCA	87.69
*cox2*	1621/1634	2245/2260	625/627	3/3	ATG/GTG	T/TAA		77.35
*nad6*	2247/2261	2693/2707	447/447	1/0	GTG/ATG	TAA/TAG		72.48
*trnL1*	2694/2708	2758/2776	65/69				TAG/TAG	78.26
*trnS2*	2759/2777	2825/2843	67/67				TGA/TGA	77.61
*trnL2*	2831/2850	2895/2914	65/65	5/6			TAA/TAA	87.69
*trnR*	2896/2916	2963/2981	68/66	0/1			TCG/TCG	72.46
*nad5*	2966/2983	4537/4548	1572/1566	2/1	ATG/ATG	TAA/TAA		68.89
*trnK*	6558/5027	6622/5091	65/65	2020/478			CTT/CTT	86.36
*trnT*	6623/5094	6686/5159	64/66	0/2			TGT/TGT	86.36
*trnW*	6689/5162	6751/5225	63/64	2/2			TCA/TCA	93.75
*trnY*	6752/5232	6814/5295	63/64	0/6			GTA/GTA	93.85
*trnS1*	6838/5301	6894/5357	57/57	23/5			GCT/GCT	82.46
*rrnL*	6895/5358	7828/6297	934/940					84.93
*rrnS*	7829/6298	8550/7030	722/733					85.56
*trnE*	8551/7031	8611/7093	61/63				TTC/TTC	77.78
*trnG*	8620/7131	8686/7196	67/66	8/37			TCC/TCC	80.6
*cox3*	9479/7613	10150/8284	672/672	792/416	ATG/ATG	TAA/TAA		76.64
*trnH*	10131/8265	10192/8328	62/64	−20/−20			GTG/GTG	89.06
*cytb*	10193/8329	11269/9405	1077/1077		ATG/ATG	TAA/TAA		84.22
*nad4L*	11269/9405	11520/9656	252/252	−1/−1	ATG/ATG	TAG/TAG		76.59
*nad4*	11608/9629	12852/10846	1245/1218	87/−28	ATG/TTG	TAG/TAA		71.73
*trnQ*	12856/10849	12916/10911	61/63	3/2			TTG/TTG	85.71
*trnF*	12915/10910	12979/10974	65/65	−2/−2			GAA/GAA	98.46
*trnM*	12971/10967	13035/11030	65/64	−9/−8			CAT/CAT	92.31
*atp6*	13039/11031	13548/11543	510/513	3/0	ATG/ATG	TAG/TAA		76.02
*nad2*	13552/11544	14373/12371	822/828	3/0	ATG/ATG	TAA/TAA		69.2
*trnV*	14378/12372	14442/12435	65/64	4/0			TAC/TAC	81.54
*trnA*	14443/12436	14506/12503	64/68				TGC/TGC	82.35
*trnD*	14506/12504	14568/12566	63/63	−1/0			GTC/GTC	81.25
*nad1*	14569/12567	15468/13466	900/900		ATG/GTG	TAA/TAA		80.56
*trnN*	15475/13468	15538/13530	64/63	6/1			GTT/GTT	84.38
*trnP*	15562/13573	15626/13639	65/67	23/42			TGG/TGG	83.82
*trnI*	15626/13639	15692/13704	67/66	−1/−1			GAT/GAT	92.54
*nad3*	15696/13711	16058/14073	363/363	3/6	ATG/ATG	TAA/TAA		75.21

**Table 2 ijms-20-04214-t002:** Pairwise common interval comparison of mitochondrial gene orders among 12 monopisthocotylean species (only one representative is shown for species with identical gene orders), based on the order of all 36 genes. Scores indicate the similarity between gene orders, where 1254 represents an identical gene order. Shading indicates the level of similarity: light to dark = similar to dissimilar.

	N	B	B	T	L	C	D	G	G	A	T	P
*Neobenedenia melleni*	1254	546	1120	294	292	1186	1056	622	1120	302	344	148
*Benedenia seriolae*	546	1254	610	162	162	580	514	342	552	184	230	84
*Benedenia hoshinai*	1120	610	1254	294	292	1186	1056	660	1120	302	356	146
*Thaparocleidus varicus*	294	162	294	1254	222	326	316	114	294	128	144	102
*Lepidotrema longipenis*	292	162	292	222	1254	322	306	110	292	146	232	182
*Cichlidogyrus sclerosus*	1186	580	1186	326	322	1254	1120	638	1186	322	370	162
*Dactylogyrus lamellatus*	1056	514	1056	316	306	1120	1254	608	1056	322	336	162
*Gyrodactylus gurleyi*	622	342	660	114	110	638	608	1254	688	252	214	94
*Gyrodactylus nyanzae*	1120	552	1120	294	292	1186	1056	688	1254	344	356	146
*Aglaiogyrodactylus forficulatus*	302	184	302	128	146	322	322	252	344	1254	150	108
*Tetraonchus monenteron*	344	230	356	144	232	370	336	214	356	150	1254	430
*Paratetraonchoides inermis*	148	84	146	102	182	162	162	94	146	108	430	1254

**Table 3 ijms-20-04214-t003:** Nucleotide composition and skewness comparison of different elements of the mitochondrial genomes of *Thaparocleidus asoti* and *Thaparocleidus varicus*. PCGs: protein-encoding genes.

Regions	Size (bp)	T(U)	C	A	G	AT(%)	GC(%)	GT(%)	AT Skew	GC Skew
*Thaparocleidus asoti*/*Thaparocleidus varicus*										
PCGs	10038/10020	48.6/49.4	7.5/7.6	26.8/26.8	17.1/16.2	75.4/76.2	24.6/23.8	65.7/65.6	−0.288/−0.297	0.393/0.364
1st codon position	3346/3340	41.7/42.3	7.8/7.2	29.7/30.5	20.8/20.0	71.4/72.8	28.6/27.2	62.5/62.3	−0.168/−0.162	0.455/0.474
2nd codon position	3346/3340	50.1/50.2	11.6/12.0	21.0/20.4	17.3/17.5	71.1/70.6	28.9/29.5	67.4/67.7	−0.410/−0.423	0.196/0.187
3rd codon position	3346/3340	53.9/55.8	3.0/3.5	29.8/29.6	13.4/11.1	83.7/85.4	16.4/14.6	67.3/66.9	−0.288/−0.308	0.631/0.516
*atp6*	510/513	50.2/49.3	6.7/8.2	27.6/26.7	15.5/15.8	77.8/76.0	22.2/24.0	65.7/65.1	−0.290/−0.297	0.398/0.317
*cox1*	1554/1557	45.3/46.6	11.0/11.0	24.8/24.0	18.9/18.4	70.1/70.6	29.9/29.4	64.2/65.0	−0.293/−0.320	0.265/0.252
*cox2*	625/627	42.4/43.1	9.3/9.7	28.2/27.8	20.2/19.5	70.6/70.9	29.5/29.2	62.6/62.6	−0.202/−0.216	0.370/0.333
*cox3*	672/672	51.3/49.9	6.5/6.8	23.5/26.2	18.6/17.1	74.8/76.1	25.1/23.9	69.9/67.0	−0.372/−0.311	0.479/0.429
*cytb*	1077/1077	47.3/47.8	8.9/8.9	25.6/26.2	18.2/17.1	72.9/74.0	27.1/26.0	65.5/64.9	−0.297/−0.292	0.342/0.314
*nad1*	900/900	48.9/49.9	8.1/7.1	26.9/27.0	16.1/16.0	75.8/76.9	24.2/23.1	65.0/65.9	−0.290/−0.298	0.330/0.385
*nad2*	822/828	51.6/54.7	5.4/5.1	28.2/26.4	14.8/13.8	79.8/81.1	20.2/18.9	66.4/68.5	−0.293/−0.348	0.470/0.462
*nad3*	363/363	49.6/49.3	3.0/5.0	30.9/29.8	16.5/16.0	80.5/79.1	19.5/21.0	66.1/65.3	−0.233/−0.247	0.690/0.526
*nad4*	1245/1218	50.8/52.6	7.5/6.7	26.5/27.0	15.3/13.6	77.3/79.6	22.8/20.3	66.1/66.2	−0.314/−0.322	0.343/0.339
*nad4L*	252/252	50.8/53.6	5.2/5.2	29.0/28.6	15.1/12.7	79.8/82.2	20.3/17.9	65.9/66.3	−0.274/−0.304	0.490/0.422
*nad5*	1572/1566	48.4/48.9	5.5/6.3	28.7/28.6	17.4/16.2	77.1/77.5	22.9/22.5	65.8/65.1	−0.256/−0.262	0.517/0.443
*nad6*	447/447	51.7/51.9	5.8/5.4	26.2/27.7	16.3/15.0	77.9/79.6	22.1/20.4	68.0/66.9	−0.328/−0.303	0.475/0.473
*rrnL*	934/940	39.8/39.3	8.5/8.4	35.1/37.0	16.6/15.3	74.9/76.3	25.1/23.7	56.4/54.6	−0.063/−0.029	0.325/0.291
*rrnS*	722/733	41.0/38.6	8.4/8.3	35.6/38.2	15.0/14.9	76.6/76.8	23.4/23.2	56.0/53.5	−0.071/−0.005	0.278/0.282
rRNAs	1656/1673	40.3/39.0	8.5/8.4	35.3/37.5	15.9/15.1	75.6/76.5	24.4/23.5	56.2/54.1	−0.066/−0.019	0.305/0.288
tRNAs	1410/1424	40.6/40.8	7.8/8.1	36.2/35.7	15.3/15.4	76.8/76.5	23.1/23.5	55.9/56.2	−0.057/−0.067	0.325/0.313
Full genome	16074/14088	46.5/46.8	7.4/7.6	31.2/30.1	14.8/15.5	77.7/76.9	22.2/23.1	61.3/62.3	−0.197/−0.217	0.334/0.341

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
