# Peer review of "Mitochondrial Genomes of Two Thaparocleidus Species (Platyhelminthes: Monogenea) Reveal the First rRNA Gene Rearrangement among the Neodermata"

_ijms, 2019, doi:10.3390/ijms20174214_

Round 1

Reviewer 1 Report

- English needs attention (although oddly, mostly in the abstract)

- it is likely that the sequencing the TRs in the NCR1 of T. asoti will require using next generation (and therefore expensive) sequencing technologies; it would be a valuable pursuit for the future, but is not necessary for the publication of this paper. If the authors choose to pursue this I recommend taking a look at Cao et al. Sequencing technologies and tools for short tandem repeat variation detection, Briefings in Bioinformatics 16(2)193-204 (2015)

- the mixture, then single specimen, sequencing is clever

- the discussion of gene order evolution is excellent

- I very much appreciate the conservative approach taken by the authors to the conclusions. We are all placed under too much pressure to **revolutionize science** with every paper, and the authors are to be commended for their resistance to such pressures.

- it is sometimes difficult to publish papers like this which present basic new data but that don’t have dramatic conclusions; these kinds of papers are vital for the establishment of robust data for comparative purposes

ln 71-72: non-represented should be unrepresented

ln 72: “data” are plural, thus “scarcity of data have so far…”

ln 279: amplified

Reviewer 2 Report

This paper describes the sequences and gene organization of mtDNAs from two species (asoti and varicus) of the genus Thaparocleidus (Platyhelminthes). The authors discuss the novel structure and gene order of the mitochondrial DNA including the strange non-coding regions (NCRs) containing highly repetitive regions (HRRs).  They use all of these features (sequence and structural variations) to argue for a specific phylogeny in this group.  The phylogeny, which is thoroughly analyzed, is the major focus of the paper, but the novel structure and its function, and the novel gene order and its evolution are of more general interest than the precise placement of these organisms within the existing phylogeny.

While the bizarre HRR structure of the NCR is discussed (Figure 2), there is no discussion or speculation about its function.  NCR regions classically include the origins of replication and promoters, and so, it is surprising that no mention was made of these possible functions through analysis of structural conservation or variation.

Also, the gene order and organization is very interesting, especially in terms of the evolution of this order, although one area of this analysis that seems very weak is the TDRL analysis and the phylogenetic conclusions.  The authors argue, based on the principle of maximum parsimony, that the putative ancestral gene order must have preceded the Thaparocleidus gene order based on the minimum number of steps (two versus three) that are needed to convert from one order to the other (Figure 5).  I think that this is a very weak argument.  First, a difference of one step is just not significant.  Evolution does not necessarily proceed along the most efficient pathway.  Second, mechanistically, these conversions must have required more steps or operations than the minimum.  It is probable that at least each set of contiguous genes recombined as a separate step.  If this is the case, then the first operation (moving counterclockwise in figure 5) would have required five steps and the second operation would have required 4 steps for a total of 9 steps in the conversion.  This also assumes that every recombination event occurred in the same direction.  The conclusion that these events were irreversible is just not correct.  I recommend that this weak argument (and figure 5) be removed from the Discussion.

Some minor recommended changes:

Line 59: there are two “the”s, one inside the parenthesis and one outside.  I recommend: It is likely that at least part of the underlying cause…..

Line 64: ………such a thing as……..  Add “a”.

Line 100:  Figure 1.  It would be nice to show the location of the HRRs within the NCRs.

Line 111:  Figure 2. The color coding in figure 2 is not defined in the legend.

Line 118:  A 129 bp ORF (43 codons) is not very significant, i.e. it could easily have happened by chance.

Line 129: BI and ML are not defined.  They are defined in the Materials and Methods (Line 319), but since the Materials and Methods follow the Results, they should be defined at first use.

Line 149: The high rate or GO rearrangements……. should be: The high rate of GO rearrangements…….
